# Recapitulating Liver Embryology—Lessons to Be Learned for Liver Diseases

**DOI:** 10.3390/jdb13040039

**Published:** 2025-11-04

**Authors:** Rui Caetano Oliveira, Sandra Ferreira, Isabel Gonçalves, Maria Fátima Martins

**Affiliations:** 1Instituto de Histologia e Embriologia, Faculdade de Medicina da Universidade de Coimbra, 3000-000 Coimbra, Portugal; 2Unidade de Hepatologia e Transplantação Hepática Pediátrica, Unidade Local de Saúde Coimbra, 3000-000 Coimbra, Portugal

**Keywords:** liver, embryology, histology, liver diseases

## Abstract

Despite looking monotonous, liver histology represents a highly complex structure of hepatocytes, bile ducts and vessels. This complex interaction and development originate in embryology and remain in adult life. In this manuscript, we highlight the features of liver embryology, translating the events into pathologic features and opening possibilities for disease understanding and research. We revisit liver embryology, from biliary to vascular processes, stressing some developing abnormalities with a focus on the histological findings. With this manuscript, we hope to increase the awareness of the importance of embryology in diseases, prompting its detailed study.

## 1. Introduction

The liver, when analyzed by histology, demonstrates a more or less monotonous histology, but it is the phenotype of a highly complex architecture of hepatocytes, bile ducts and blood vessels that is not fully understood [1,2].

The purpose of this work is to review liver embryology, highlighting specific features that are important to the development of liver diseases, namely developmental anomalies. Other liver pathologies such as tumors or cirrhosis are not covered by this manuscript.

Histology is pivotal is establishing liver diseases’ etiology and may have a prognostic role by assessing features of disease, both in adults [3] and in children [4]. Reference centers in liver pathology, especially pediatric ones, perform their diagnostic algorithms with at least highly recommended liver biopsies. In addition, liver biopsy is also a valuable method for ancillary testing, such as immunohistochemical and molecular studies [5].

The knowledge of liver histology is therefore a plus in understanding pathologies, especially for development diseases.

Liver is a unique organ in the human body, in a central position, responsible for metabolic activities and with unique anatomic relations, as seen in Figure 1.

This review was conducted with a generic method. We conducted a search in indexed databases, such as PubMed and MedLine, using the keywords “liver embryology”, “bile duct embryology”, “liver formation”, “bile duct formation”, “liver development”, “liver embryology genetic regulation” and “bile duct development”. Only articles fully available in English were considered, from 2000 and forward. This research started in December 2022 and was finished in December 2024.

## 2. Hepatic Embryology

### 2.1. Liver Development

Considering human embryology, the first sign of liver development arises at 3 weeks and 7 days. At this point, there is a thickening of the endodermal cells in the embryonic foregut, giving birth to the hepatic diverticulum [6]. These cells, under the influence of mesodermal signals, expand and form cords from the cephalic part of the hepatic diverticulum into the septum transversum, which anastomose around pre-existing endothelial spaces, thus becoming more organized, while the septum transversum becomes the liver capsule [7]. These cells constitute the majority of the liver mass and are called hepatocytes and are held together by tight junctions and have a basolateral membrane facing the sinusoids (where they release their endocrine secretions) and an apical pole towards the bile canaliculi, which posteriorly merge in the portal bile duct [8]. The hepatocytes have a central role in liver function, and very interestingly there is an uneven function distribution along the porto-central axis in the liver cell plate, designated as “liver zonation” [9]. This event is regulated by the Wnt/β-catenin developmental pathway, a key player in liver development and pathogenesis [10]. In addition to being crucial in the proliferation and differentiation of hepatic progenitor cells (hepatoblasts to form hepatocytes and cholangiocytes), it is a seal of various hepatic pathologies [11]. This pathway is activated in pericentral hepatocytes, inducing a specific program which depends of the absence of the adenomatous polyposis coli (APC) tumor suppressor gene [11]. In the periportal area, the presence of the APC gene negatively regulates the Wnt signaling, favoring the Ras/MAPK/ERK signaling pathway’s activation, resulting in the periportal phenotype [12]. This metabolic zonation, a fundamental process in liver development and function, has been linked to different tumor morphological subtypes and prognoses [13].

### 2.2. Biliary Duct Development

The intrahepatic bile ducts start their development at the moment that the primitive hepatocytes come into contact with the mesenchyme that composes the portal tract [14].

This process can be divided into five steps [2]: (step 1) between the 13th and 14th embryonic day, a subclass of primitive hepatocytes that express biliary cytokeratins (the remaining hepatocytes express lower levels of these keratins) can be identified around the portal tract mesenchyme and are denominated as biliary precursor cells; (step 2) the second step initiates on the 15th day, where the biliary precursor cells are distributed circumferentially around the portal mesenchyme and continuously in a single layer, thus being called the ductal plate; (step 3) in the third step, around the 16th day, there is a duplication of that single layer; and (step 4) at the 17th day, the ductal plate experiences a major modification, and focal dilations arise between the double layer, constituting ducts. The portions of the ductal plate not involved in this process are marked for regression, leaving only the biliary ducts, which in the final stage (step 5), around birth, are incorporated into the portal tracts [15]. An example of ductal plate development can be seen in Figure 2 and Figure 3.

This process, which seems simple, is mediated by complex mechanisms not totally understood. Some studies point out that the protein hepatocyte nuclear factor (HNF)-6 is an initial and key player in this process since it is highly expressed in hepatoblasts and biliary precursor cells, but not in the mesenchyme [16]. The portal mesenchyme is also fundamental, since it stimulates the differentiation of hepatoblasts into biliary precursor cells by FoxF1 modulation, controlling the biliary precursor cells’ development, which in turn controls the mesenchyme development through very interesting crosstalk [17]. Other components of the extracellular matrix have been identified as important in this process, such as laminin, fibronectin and collagen type I and IV, among others [18]. A major key player in this cholangiocyte development is the Notch signaling pathway. This pathway is an evolutionarily conserved mechanism, responsible (among other functions) for mediating cell proliferation, the maintenance of progenitor cell/stem population and biliary duct formation [19].

The interaction between hepatocytes, biliary cells and the mesenchyme is summarized in Figure 4.

An interesting question of the bile duct embryology is the mechanism by which the single layer becomes a double layer, namely the transition from step 2 to step 3. Initially thought to be a simple division of the single layer, the proliferative index of the single layer was found to be not high enough [16], and the proliferative activity is only found later when the bile ducts are incorporated [18] in the portal mesenchyme. A valid option is that the second layer originated from a differentiation of hepatoblasts in biliary precursor cells [21]. This has to be a different mechanism of the first layer, since now there is a barrier of cells between hepatoblasts and the portal mesenchyme, resulting in a very different microenvironment [22].

The regression of step 4 is mainly mediated by apoptosis, with studies finding diminished levels of Bcl-2 (an anti-apoptotic) in stage 4 compared to stage 2 [23].

In stage 4–5, it is common to consider that the ducts “invade” the portal mesenchyme, but some studies consider that it is the mesenchyme that envelops the ducts, once again in a very coordinated crosstalk between cells and the matrix, with tissue remodeling and tubulogenesis [24].

This process is regulated at the molecular level. The cells of the ductal plate, near the portal tracts, have an expression of sex-determining region Y box 9 (Sox9) [25], while those facing the hepatic parenchyma have an expression of Hepatocyte Nuclear Factor (HNF) 4α [26]. In practice, this means that the hepatocytes that are not destined to become bile ducts have a repressed biliary phenotype via the TGFβ signaling pathway and differentiate into hepatocytes [27]. The contact with the portal mesenchyme induces biliary differentiation. The portal mesenchyme expresses laminin and Jagged1 [28], which interact with Notch2 and prompt this differentiation [29,30]. This is a rather unique phenomenon, and it is supported by the fact that there is no biliary differentiation around the hepatic central vein. The proper development of the intrahepatic biliary tree depends on the Notch signaling and is evidenced not only in animal models with decreased Notch signaling but also in human patients with Alagille syndrome [31].

The connection of the newly formed ducts, ductal plate and biliary canaliculi is influenced and maintained by constant tissue remodeling under the influence of three major genes: WNT, TGFβ and Notch [32,33]. This process of remodeling is still present at birth, where it reaches the smallest and peripheral portal tracts, and some ductal plates may persist. This process begins at the hilum, around the largest portal tracts, progressing outwards [34].

Biliary canaliculi are evident before the twelve-week gestation point and show up as intercellular spaces between adjacent hepatocytes [15].

The gallbladder and extrahepatic bile ducts are not the main subjects of this review, but they develop differently. As we say, the intrahepatic bile ducts are developed simultaneously and intimately with the liver, while the gallbladder and extrahepatic bile ducts arise from the caudal portion of the hepatic diverticulum [18]. So, they originate directly from the endoderm, sharing a common origin with the ventral pancreas (the intrahepatic biliary ducts derive from hepatoblasts at the ductal plate) [35].

As the duodenum pulls out from the septum transversum, the stalk of the hepatic bud gives birth to the extrahepatic ducts, which is continuous from the caudal extremity to the duodenum and at the cephalic pole with the primitive hepatic sheets [36]. This cephalic extremity provides the origin for the hepatic ducts and part of the cystic duct [37]; the caudal extremity is responsible for forming the gallbladder, part of the cystic duct and the common bile duct [7].

### 2.3. Vascular Development

The hepatoblasts that invade the mesoderm surround the pre-existing veins, thus giving birth to the early sinusoids [38]. Blood supply is provided initially by the vitelline veins and in a later phase is provided by the left and right umbilical veins, both rich in oxygen and nutrients from the placenta [39]. Posteriorly the right umbilical vein fades away, leaving the left umbilical vein as the main blood supplier to the liver. The blood from the left umbilical vein then goes to the sinusoids of the left side of the liver, to the sinusoids of the right side of the liver and to the inferior vena cava via the ductus venosus [40]. Nowadays it is established, by radiological studies (ultrasound), that the blood supply of the left lobe is almost all provided by a nutrient-rich umbilical vein, while the right lobe has a 50/50 supply from the umbilical vein and the portal vein (this a nutrient-poor vein) [41]. As a consequence, the left lobe of the liver is better perfused in the uterus and, thus, able to better withstand hypoxic injuries [42,43]. At birth, the umbilical vein transforms into the ligamentum teres and the ductus venosus in the ligamentum venosum [44].

Regarding the hepatic artery branches, they are evident in the developing portal tracts (first near the hilum and later in the periphery) in a spatial and temporal development that mimics the developing bile ducts [38]. This is thought to be induced by the secretion of the vascular endothelial growth factor (VEGF) by the ductal plate [45].

## 3. Abnormalities of Development

As exposed liver development is a highly complex process and is occasionally prone to abnormalities in its development, these abnormalities may be classified according to their main mechanism (hepatic, vascular or biliary). We will focus on these three groups and discuss some of the main features, including histological ones.

### 3.1. Hepatic Abnormalities

In this group, we have considered the agenesis. Total liver agenesis is very rare and not compatible with life. Hepatic lobe agenesis is also rare, and in this abnormality, a right liver lobe agenesis has been reported more often than a left liver lobe agenesis [46,47].

Abnormalities in the liver position are more frequently reported. In situs inversus, the liver is present on the left side of the abdominal cavity and may be associated with biliary atresia [48].

Biliary atresia (BA) is a rare disease that may be characterized as an idiopathic obliterative cholangiopathy that affects both intra- and extrahepatic bile ducts, leading to progressive liver fibrosis and cirrhosis if left untreated [49]. It is responsible for the highest percentage of pediatric liver transplants, presenting in the newborn period with conjugated jaundice, pale stools and dark urine. The diagnosis is based on a combination of clinical manifestations, laboratory tests and imaging findings (on abdominal ultrasound). A liver biopsy supports the diagnosis, but intraoperative cholangiogram provides the direct visualization of the biliary anatomy and is the most accurate diagnostic test. It is typically performed during the Kasai procedure (hepatoportoenterostomy), which is the surgical treatment for BA. An early intervention, (ideally before 60 days of life), significantly improves outcomes [50]. The pathophysiology of BA remains unclear, and BA is not a single homogeneous disease but rather the final common outcome of multiple distinct etiological mechanisms (e.g., genetic, epigenetic or virus-induced).

Liver biopsy has a pivotal role in classifying the severity of the disease, especially the porta hepatis liver biopsy, and exhibits morphological findings that support the concept of disordered embryogenesis [51]. The evaluation of the porta hepatis allows for the assessment of the size of the bile ductules, thus predicting the possibilities of the long-term outcome of the Kasai portoenterostomy [52] and consequently being an important element in the decision for liver transplants. A summary of histological changes can be seen in Figure 5.

Although BA is usually considered as a standalone anomaly in otherwise normal infants, a minority of patients present with extrahepatic congenital anomalies, which in some instances are classified as syndromes—the most well-characterized being Biliary Atresia Splenic Malformation (BASM). The key components of BASM include a characteristic spectrum of anomalies, such as situs inversus, intestinal malrotation or non-rotation, polysplenia or asplenia, the absence of the inferior vena cava, preduodenal portal veins and cardiovascular abnormalities [49,50]. Approximately 30–50% of patients present with situs inversus and are cited as examples of “laterality defects”, supporting an origin during early embryonic development [50].

In cases of diaphragmatic hernias or omphaloceles, the liver may be displaced and protrude into the thoracic cavity [54,55], as demonstrated in Figure 6.

The etiology of congenital diaphragmatic hernias is not totally understood, and it is thought to be multifactorial [56]. Multiple genes have been associated with it, with the most common conditions being trisomy 18, trisomy 13, trisomy 21 and Turner syndrome (45, X); rarer conditions such as Fryns syndrome and Cornelia de Lange syndrome have been described [57]. Exposure during pregnancy to teratogenic agents such as mycophenolate mofetil, allopurinol and lithium has also been linked to congenital diaphragmatic hernias [56], and some authors have stated the role of metabolic disturbances in the vitamin A pathway [58].

Accessory lobes are rather frequent and usually arise from the inferior liver surface and sometimes may form a tongue-like projection in the right lobe—Riedel’s lobe [59].

Ectopic liver tissue is also described and may be detected in the liver suspensory ligaments, wall of the gallbladder, splenic capsule, greater omentum and retroperitoneal space [60]. It is a rare developmental condition often discovered incidentally (imaging, laparotomy, laparoscopy or during an autopsy) [61]. While it is predominantly asymptomatic, rare cases have reported recurrent abdominal pain due to torsion, intraperitoneal bleeding, hemorrhagic necrosis and the compression of adjacent organs, as well as the obstruction of the esophagus, portal vein and neonatal gastric outlet [61]. Multiple studies have documented cases of hepatocellular carcinoma arising from ectopic liver tissue [61]. Although rare, the potential increased risk of malignant transformation in ectopic liver tissue warrants careful management. The excision of ectopic liver tissue followed by a histopathological examination is recommended [62].

Heterotopia in the liver is also reported: adrenal tissue in the liver is usually present in cases of hepatoadrenal fusion (lack of adrenal capsule), and pancreatic ectopic tissue can be frequently found in the hepatic hilum [63,64], as seen in Figure 7.

### 3.2. Vascular Abnormalities

A disturbance in portal vein development may lead to its absence, a preduodenal position, cavernous transformation (so-called cavernoma), duplication and congenital portosystemic communications [65,66]. Congenital portosystemic shunts (CPSSs) are rare vascular malformations of an embryonic origin in which the blood flow coming from the intestine bypasses the liver, either partially or completely, reaching the systemic circulation unfiltered. CPSSs—Abernethy Malformation [67,68]—can be intrahepatic or extrahepatic and are usually classified into type I—the venous portal blood is completely diverted from the portal vein to the inferior vena cava, bypassing the liver—and type II—only part of the blood is diverted to the inferior vena cava. Type I shunts are more frequent in girls and may be associated with other abnormalities—such as liver atresia and liver tumors. Type II shunts are less common and usually are detected later during medical investigation due to encephalopathy.

Aberneathy Malformation is a rare condition with a probably understated incidence due to the lack of standard protocol. The (probably) most common cohort of these patients [69] described a rather high risk of complications, such as hepatic encephalopathy and liver lesions. CPSSs are indeed associated with a broad spectrum of complications, most notably pulmonary issues (portopulmonary hypertension and hepatopulmonary syndrome), neurological complications (neurocognitive dysfunction and encephalopathy) and endocrine/metabolic issues (hypothyroidism, hyperinsulinaemic hypoglycemia, impact on linear growth, hyperandrogenism), as well as cholestasis/hyperbilirubinemia and a high prevalence of liver nodules. The majority of hepatic nodules are benign, such as focal nodular hyperplasia and hepatocellular adenoma, but hepatocellular carcinoma has also been reported more frequently in adolescents and adults. The evaluation of hepatic nodules in the context of CPSSs is particularly challenging, as their radiological, histological and molecular features, as well as their risk of malignant transformation, differ from those of lesions arising in other settings. Moreover, the current histopathological classification may not always be applicable [70].

The histological picture is rather variable but includes the absence of portal veins in smaller portal tracts and hypoplastic (or absent) veins in medium- and large-sized portal tracts; the remodeling of the liver architecture is also evident with nodular regenerative hyperplasia and isolated capillaries and arterioles in the lobules [71]. This malformation is commonly associated with liver lesions, including hepatocellular carcinomas [72].

Due to the risk of liver nodules, especially hepatocellular carcinomas, a liver biopsy is recommended as part of the treatment algorithm and may be fundamental for decisions to perform a liver transplant [73,74]. 

A histologic representation of the Abernethy Malformation in the liver is depicted in Figure 8.

CPSSs are diagnosed by Doppler ultrasound, contrast-enhanced ultrasound or cross-sectional imaging (angiography, hepatobiliary contrast-enhanced MRI). The preferred imaging baseline is contrast-enhanced. However, the most accurate method to confirm and characterize the shunt anatomy is phlebography with an occlusion test [70]. In addition to the specific evaluation and supportive treatment of each complication, endovascular shunt occlusion is generally recommended, after assessing contraindications to closure. Recently, following a symposium on this topic, a multidisciplinary group of recognized experts published the following recommendations on behalf of the European Association for the Study of the Liver (EASL) [70]: (1) follow all asymptomatic intrahepatic CPSSs detected at birth longitudinally until spontaneous closure and 1 year beyond documented closure; (2) close asymptomatic intrahepatic CPSSs if they do not close spontaneously within the first 2 years of life; (3) close all asymptomatic extrahepatic CPSSs pre-emptively, as early as possible; (4) close all symptomatic CPSSs beyond the neonatal period; and (5) in cases of prenatally detected CPSSs, perform prenatal and neonatal assessments in a specialized center, since the closure of an extrahepatic CPSS may be indicated early in the postnatal period.

Hepatoportal arteriovenous connections can also develop and may be secondary to trauma or biopsy and even congenital as part of a syndrome like Osler–Weber–Rendu syndrome [75,76,77]. These connections can lead to portal hypertension and induced lesions in the spectrum of non-cirrhotic portal hypertension, as seen in Figure 9.

Non-cirrhotic portal hypertension (NCPH) refers to a subset of conditions in which portal hypertension develops in the absence of cirrhosis. It has been reported in association with immune disorders (autoimmune or immunodeficiency), infections, malignancies, prothrombotic states, cardiocirculatory disorders, certain genetic syndromes (e.g., Turner, Adams–Oliver and Felty syndromes) and exposure to specific medications or toxins [78]. The pathogenesis of NCPH is complex and remains incompletely understood. It appears to represent a final common pathway resulting from various intrahepatic histologic vascular alterations. Notably, similar histopathological changes, primarily involving the hepatic sinusoids and (peri)portal vasculature, have also been observed in patients without clinical evidence of portal hypertension. In light of this, the term “porto-sinusoidal vascular disorder” (PSVD) has recently been proposed to encompass this broader spectrum of patients. NCPH is part of PSVD. Furthermore, the PSVD definition includes individuals with portal vein thrombosis and/or chronic liver disease, provided that cirrhosis is absent [78]. A liver biopsy is mandatory for the diagnosis. Specific histological findings of PSVD are as follows: obliterative portal venopathy/portal vein stenosis, nodular regenerative hyperplasia (best demonstrated with a reticulin stain) and incomplete septal fibrosis.

Patients are mostly asymptomatic until complications of portal hypertension develop. Occasionally, elevations in liver enzymes may occur. Hepatic encephalopathy or jaundice are rare [78]. The clinical course of patients with PSVD without portal hypertension remains incompletely understood. Prospective follow-up studies are needed to better define the natural history in these cases (e.g., platform at the ERN RARE-LIVER, European Reference Network for rare liver diseases).

### 3.3. Biliary Abnormalities

Structural abnormalities are rather common and consist mainly of duplications, accessory ducts or anomalous insertions of the gallbladder, hepatic and cystic ducts [79,80]. These events are thought to originate due to the abnormal remodeling of the caudal end of the hepatic bud.

Probably the most important anomaly to be recognized is the ductal plate malformation (DPM). This anomaly represents a failure in the biliary remodeling process, which leads to a persistence of the ductal plate structures [81]. All levels of the biliary tree are susceptible to this malformation. The timing of the event will determine the portion of the biliary tree that is affected. An early insult will affect ducts of all calibers, while a later insult will mainly affect small biliary ducts [82]. Thus, DPM includes congenital hepatic fibrosis, Caroli’s disease, the Von Meyenburg complex and polycystic liver disease [83].

Animal models of DPM have been developed and are the genetic basis of the DPM classification [84,85]: deficiency in transcription factor HNF6; deficiency in liver-specific transcription factor HNF1β; and deficiency in cistin-1.

The deficiency in transcription factor HNF6 [86] is associated with a defective differentiation of biliary precursors—type 1; the deficiency in liver-specific transcription factor HNF1β [87] shows a failed maturation (or asymmetric maturation) of the primitive ductal structures in the ductal plate—type 2; and the deficiency in cistin-1 is responsible for a phenotype of abnormal ductal expansion—type 3 [84].

These changes will result in altered signaling of the Notch, TGFβ and WNT pathways and, consequently, modifications in several proteins, including the ones responsible for the primary cilium [88]. Therefore, there is an inadequate response to mechanical, chemical and osmotic signals, promoting cyst formation. The importance of the cilia’s integrity and their dysfunction has prompted the name “ciliopathies” [89,90]. Primary cilia are organelles that play a pivotal role as mechano-, osmo- and chemo-sensors, as well as in signal transduction. They are essential for maintaining cell polarity, establishing the basal pole and facilitating intercellular communication. These functions are critical for ensuring tissue homeostasis, coordinated growth and structural organization. Genetically determined defects in primary cilia can impair the embryonic development of various organs, including the liver—where they are expressed in cholangiocytes—primarily leading to ductal plate malformations. Clinically, these anomalies often present as hepatic cysts and/or congenital fibrosis.

Histological changes usually include the presence of numerus ductules at the margins of portal tracts with slit-like or dilated lumens, which can anastomose and contain inspissated bile, supported by a dense and hyaline stroma [82], as demonstrated in Figure 10 and Figure 11. The ducts can be highlighted by CK7 immunostaining—Figure 12.

Fibrocystic changes can occur only in the liver but arise mainly in conjunction with other organs such as the kidney, skeletal system or central nervous system [91]. Since the morphology of these diseases has a great morphological overlap, the classification is moving towards a molecular classification.

In recent years, the genetic basis of this complex and heterogeneous group of conditions has been identified, with growing insight into genotype–phenotype correlations, particularly concerning onset age, clinical presentation and severity and disease progression [84,92].

In adults, this group of disorders is typically inherited in an autosomal dominant pattern, with cyst formation being the predominant feature. In contrast, pediatric cases more commonly follow an autosomal recessive inheritance, where hepatic fibrosis is the leading manifestation. In both forms, associated renal involvement, manifesting as either cystic or fibrosis, is often observed [83].

The autosomal dominant forms seen in adults are classified into two types based on the main features: autosomal dominant polycystic kidney disease (ADPKD) and autosomal dominant polycystic liver disease (ADPLD) [93]. ADPKD is primarily caused by mutations in the PKD1 (80%) or PKD2 (20%) genes, which generate a mutated protein product called polycystin 1, a mechanoreceptor in the cilium [92], leading to the formation of renal cysts and, frequently, extrarenal manifestations such as hepatic cysts. It is the most common form of polycystic kidney disease, and about a half of patients progress to end-stage renal disease by the sixth decade of life [83,94]. In contrast, renal function remains normal in ADPLD, which is characterized by the progressive development and enlargement of hepatic cysts that are not connected to the biliary tree. ADPLD is associated with mutations in various genes, such as PRKCSH and SEC63, which are involved in protein processing within the endoplasmic reticulum.

Unlike the adult forms, recessive fibrocystic liver diseases in children are characterized by cystic dilatations that are continuous in the biliary tree. Progressive portal tract fibrosis may occur along a spectrum, ranging from interlobular bile duct dilatation, as seen in congenital hepatic fibrosis (CHF), to the dilatation of the large intrahepatic bile ducts, characteristic of Caroli disease. When both large and small bile ducts are involved, the condition is referred to as Caroli syndrome.

CHF typically occurs in patients who also have manifestations in other organs, particularly the kidneys, as seen in autosomal recessive polycystic kidney disease (ARPKD), which is the most known and studied of this group, due to mutations in the PKHD1 gene (encodes fibrocystin located on the membrane of the cilia), generating a mutated polycistin 2 and a calcium channel, thus increasing the amount of intracellular calcium and cAMP [95].

Recently, mutations in the DZIP1L gene, which encodes a protein involved in the ciliary transition zone, have been identified as a cause of autosomal recessive polycystic kidney disease, expanding the phenotypic and genotypic spectrum of ciliopathies [96]. When bi-allelic truncating mutations are present, kidney disease is typically severe and manifests prenatally or in the neonatal period, with enlarged, echogenic kidneys as a hallmark feature. Approximately 50% of surviving children progress to kidney failure. In contrast, the genotype–phenotype correlation becomes less evident in the presence of milder mutations. In these cases, renal enlargement and dysfunction may be subtle, and hepatic complications related to congenital hepatic fibrosis (CHF) often predominate [83].

Several genes have been identified as causing congenital hepatic fibrosis (CHF) in association with the involvement of other organs—such as the brain, eyes and pancreas—defining a range of syndromic forms (e.g., CHF is not a distinct, isolated entity but rather part of a broad spectrum of disorders).

Patients with isolated hepatic involvement are frequently asymptomatic; however, some may experience intracystic hemorrhage and/or infection, portal hypertension or symptoms due to the mass effect of enlarging cysts. In rare cases, there is a potential risk of malignant transformation [83,93].

Ultrasound should be the first imaging modality used to diagnose simple hepatic cysts and PLD. The number of lesions (solitary vs. multiple) and the architecture (simple vs. complex cyst) are key elements in the description of hepatic cyst(s) [93].

As PLD typically does not impair liver function, treatment focuses on symptom relief. Intracystic hemorrhage usually resolves spontaneously. Cyst infections are generally treated with antibiotics, though drainage may be necessary in some cases. Treatment decisions are symptom-driven but tailored according to the number, size and location of liver cysts, along with the expertise available at the treating center.

Patients PLD should be considered for liver transplantation when the disease causes significant symptoms that severely reduce quality of life, or when complications arise that cannot be managed by any treatment other than transplantation. Such complications include severe malnutrition, the obstruction of hepatic venous outflow, ascites, portal hypertension, variceal bleeding and recurrent liver cyst infections. Transplantation is also appropriate when non-surgical treatments have failed or cannot be performed. If kidney function is also severely impaired, a referral for combined liver–kidney transplantation should be considered [93].

Biliary cysts are rather common [97]. At least some simple and solitary bile duct cysts are developmental in origin, even if present only in adulthood, and are thought to originate in progenitor cells of the ductal plate. From a histological standpoint they are rather simple, with a single layer of biliary epithelial cells and a fibrous wall.

The congenital cysts also include the ciliated hepatic foregut cyst that may arise from the incorporation of foregut structures into the septum transversum [98]. This is the only ciliated cyst of the liver and is rather straightforward to diagnose with four layers: a ciliated pseudostratified columnar epithelium, subepithelial connective tissue, a layer of smooth muscle and a fibrous capsule [99].

In the categories of biliary cysts, a well-known one is the choledochal cyst [100]. However, this designation encompasses a heterogeneous group of conditions that present dilations of several parts of the biliary tree—both intra- and extrahepatic [101]. They are more common in girls and in the oriental population [97].

Classically five types of cysts have been described (Todani classification) [102,103]: type I (the most common—circa 70%) is a cystic or fusiform dilation of the common bile duct, followed by the type Iva—multiple dilations of the intra- and extrahepatic ducts; types II (diverticulum), III (choledochocele), IVb (multiple extrahepatic cysts) and V (multiple intrahepatic cysts) are rare. Type V is the same as Caroli disease.

The etiology of these cysts has been the object of discussion: some state they are derived from an acquired weakness of the bile duct wall secondary to the reflux of pancreatic secretions [104], sometimes aggravated by an anomalous pancreatobiliary junction [105,106].

As seen, liver development is a highly complex and regulated process. Diseases developing from developmental abnormalities can exhibit a wide spectrum and are summarized in Table 1.

## 4. Conclusions

In short, knowledge of liver embryology is fundamental for understanding the basis of liver diseases. Histological changes are a phenotypic manifestation of the disease, and their interpretation is a valuable tool for disease assessment, as well as developing treatment options

## Figures and Tables

**Figure 1 jdb-13-00039-f001:**
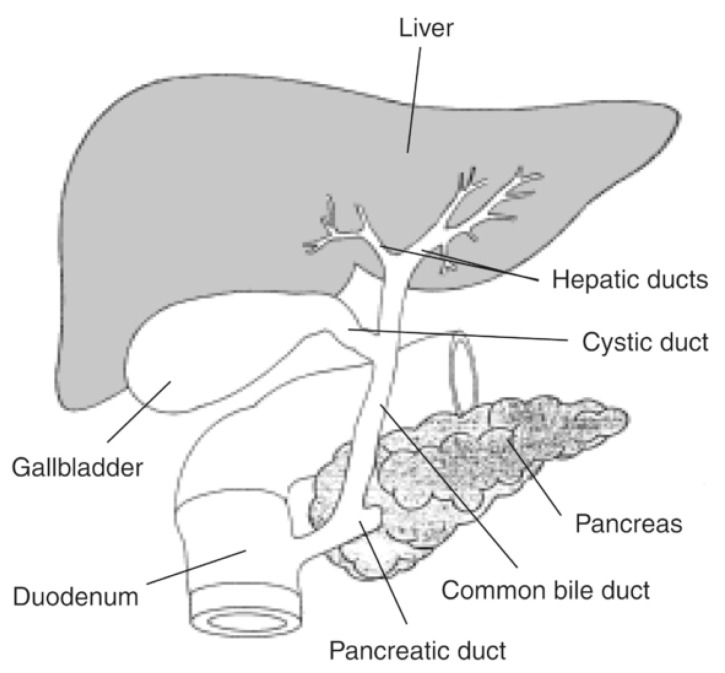
Liver and bile ducts’ anatomical relations. Available at National Institute of Diabetes and Digestive and Kidney Diseases, National Institutes of Health (https://www.niddk.nih.gov/news/media-library/17492, accessed in 27 October 2025).

**Figure 2 jdb-13-00039-f002:**
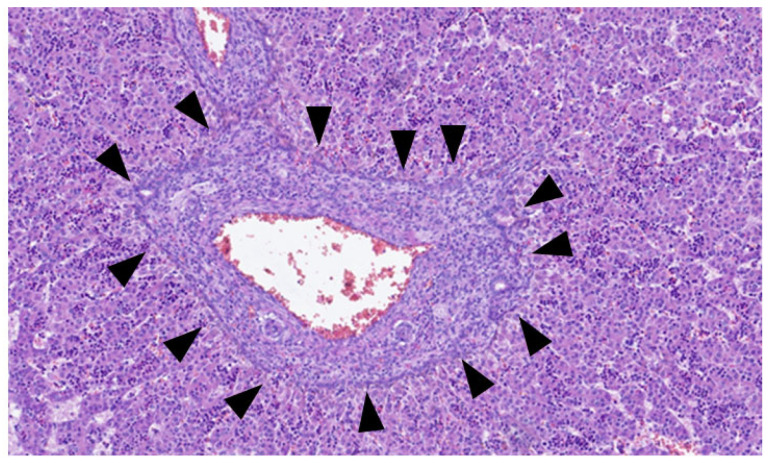
Portal tract with ductal plate (black triangles)—fetus at 16 weeks.

**Figure 3 jdb-13-00039-f003:**
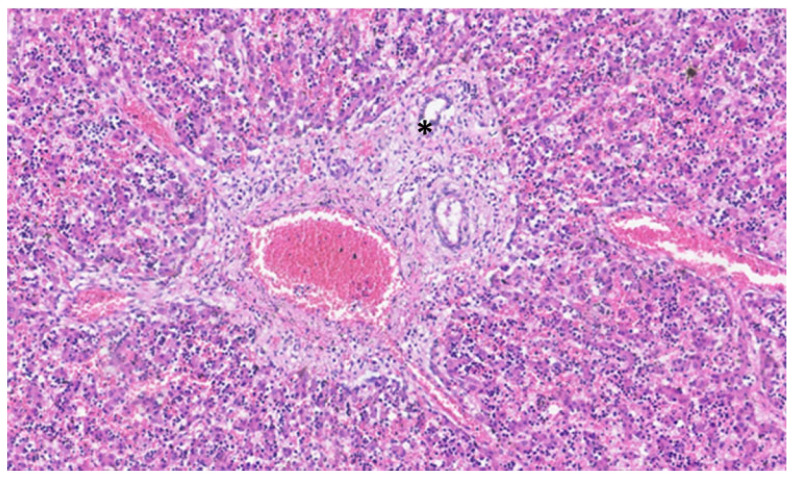
In this liver of a neonate, the ductal plate is no longer evident, and a central biliary duct (asterisk) is evident in the portal tract.

**Figure 4 jdb-13-00039-f004:**
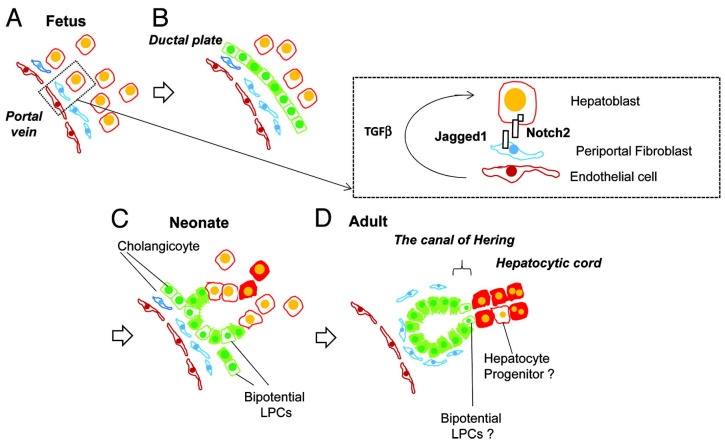
Liver development and transition of tissue localization of liver progenitor cells. Hepatoblasts are the abundant cells in fetal liver (**A**), which differentiate into cholangiocytes due to Notch signaling pathways’ influence, via interaction with Jagged-1 mesenchymal fibroblasts and TGFβ influence (**B**). These cholangiocytes have the ability to work as liver progenitor cells and can differentiate into hepatocytes (**C**) and in the adult liver are usually located in the canal of Hering. (**D**) from Tanimizu, N. & Mitaka, T. (2014). Re-evaluation of liver stem/progenitor cells. Organogenesis, 10(2), 208–215. https://doi.org/10.4161/org.27591 [20]. Reproduction allowed under the terms and conditions of the Creative Commons Attribution (CC BY) license.

**Figure 5 jdb-13-00039-f005:**
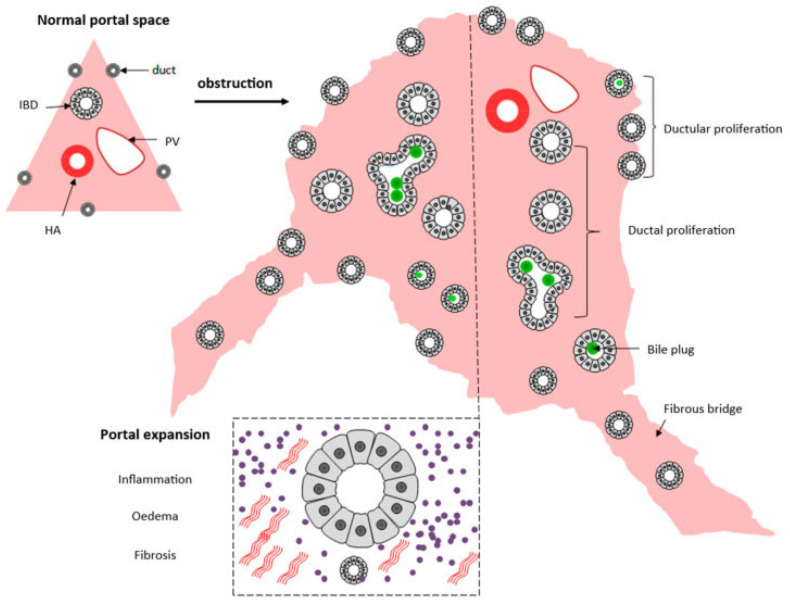
Schematic presentation of the histopathological extrahepatic obstructive histological pattern. From [53]. Reproduction allowed under the terms and conditions of the Creative Commons Attribution (CC BY) license.

**Figure 6 jdb-13-00039-f006:**
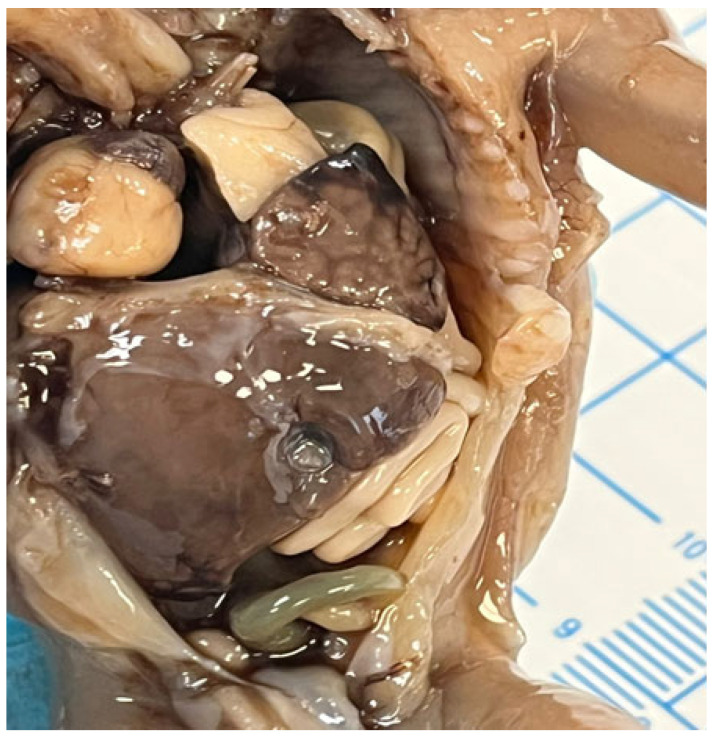
Left diaphragmatic hernia with left liver lobe protrusion into the thorax.

**Figure 7 jdb-13-00039-f007:**
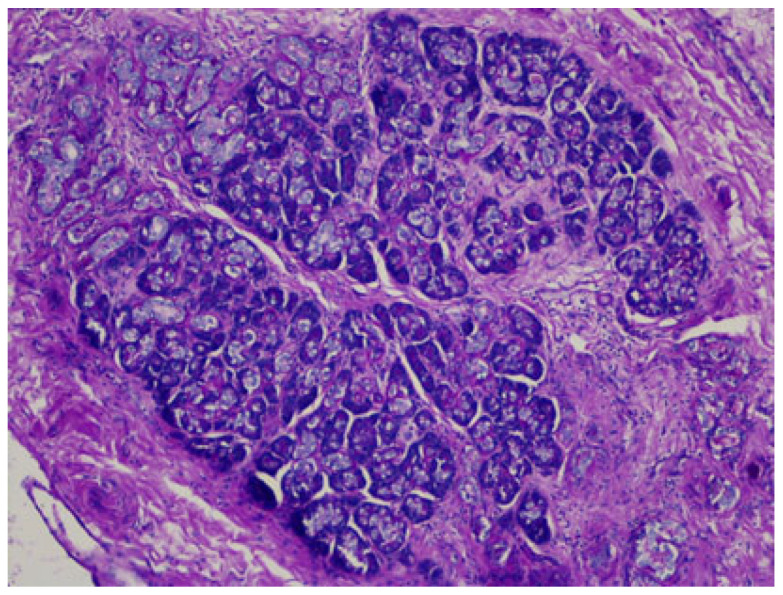
Pancreatic heterotopia in the hepatic hilum, DPAS staining.

**Figure 8 jdb-13-00039-f008:**
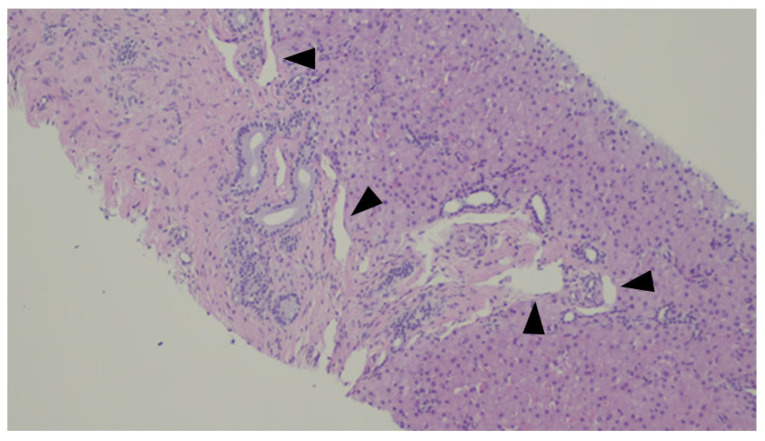
Liver biopsy of a patient with Abernethy Malformation. It is possible to see abnormal portal veins in a large portal tract. The veins are dislodged to the periphery of the tract and may herniate into the liver parenchyma (black triangles).

**Figure 9 jdb-13-00039-f009:**
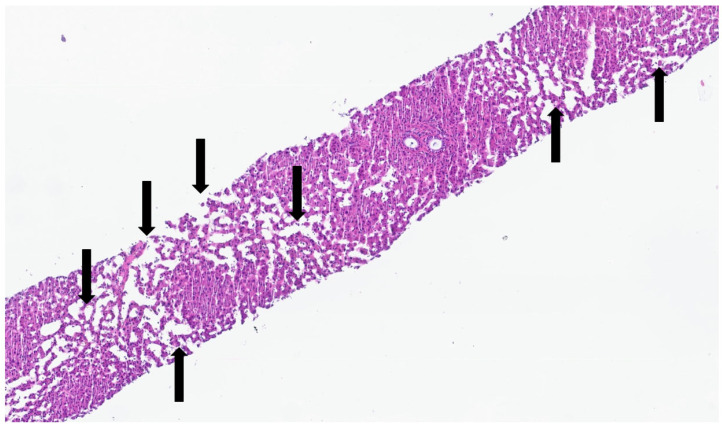
Liver biopsy of patient with Olser–Weber–Rendu syndrome—there is diffuse sinusoidal dilation (black arrows). The patient had non-cirrhotic portal hypertension.

**Figure 10 jdb-13-00039-f010:**
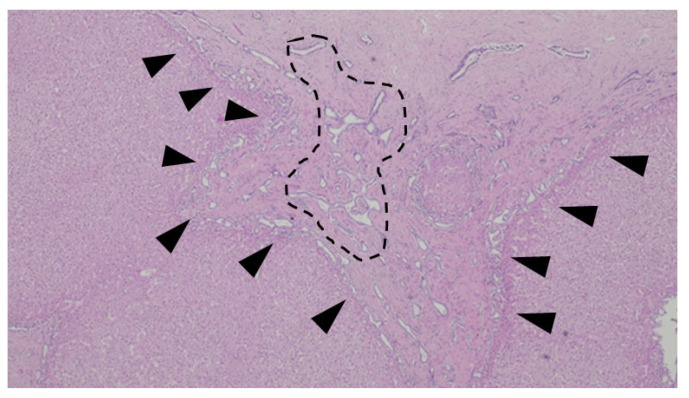
An example of ductal plate malformation—anastomosing bile ducts, sometimes dilated (dotted line), in a fibrous stroma with peripheral ductal plate-like proliferation (black triangles).

**Figure 11 jdb-13-00039-f011:**
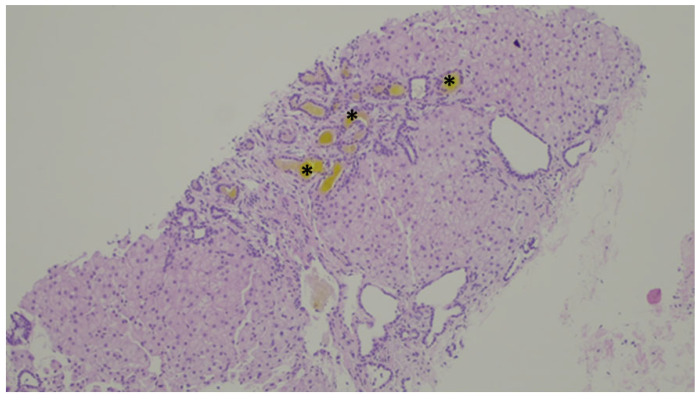
Inspissated bile ducts (asterisk).

**Figure 12 jdb-13-00039-f012:**
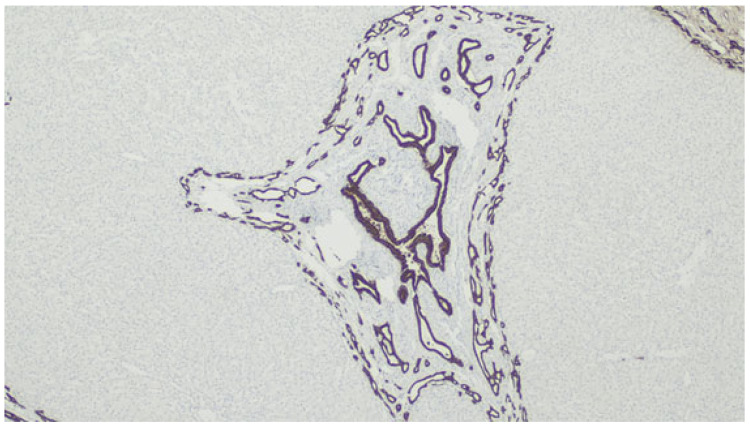
Immunolabeling with CK7 highlighting the biliary structures.

**Table 1 jdb-13-00039-t001:** Summary of the lesions described in this manuscript, their mechanism and clinical relevance.

Lesion	Mechanism Group	Frequency	Clinical Relevance
Accessory hepatic lobe/Riedel’s lobe	Hepatic	Rather frequent	Usually incidental; important mainly in surgery/imaging
Cavernous transformation of portal vein (“cavernoma”)	Vascular	Developmental anomaly; not framed as rare	May be incidental or cause mild portal HTN
Congenital portosystemic shunts (CPSS/Abernethy)	Vascular	Strong emphasis, clinically critical, underdiagnosed	Neuro- and cardio-pulmonary and high HCC risk
PSVD/NCPH spectrum	Vascular	Relevant, requires biopsy	Non-cirrhotic portal hypertension; needs follow-up
Biliary atresia	Hepatic/biliary positional defect	Rare overall but most common cause of pediatric liver transplant	Severe neonatal cholestasis; urgent surgical condition
Structural biliary variants (duplications, accessory ducts, etc.)	Biliary	Common	Critical during surgery (risk of iatrogenic injury)
Ductal plate malformations (CHF, Caroli)	Biliary/ciliopathy	One of the most important anomalies to recognize	From benign to severe portal HTN/transplant-level
Polycystic liver disease (ADPLD/ADPKD expression)	Biliary/ciliopathy	Very frequent extrarenal hepatic manifestation	Mass effect, infection, sometimes transplant required

HTN—hypertension; HCC—hepatocellular carcinoma; CPSS—congenital portosystemic shunts; PSVD—porto-sinusoidal vascular disease; NCPH—non-cirrhotic portal hypertension; CHF—congenital hepatic fibrosis; and ADPLD/ADPKD—autosomal dominant polycystic liver disease and autosomal dominant polycystic kidney disease.

## Data Availability

All data is included in the manuscript.

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
