# Peer review of "Recapitulating Liver Embryology—Lessons to Be Learned for Liver Diseases"

_jdb, 2025, doi:10.3390/jdb13040039_

Round 1
Reviewer 1 Report
Comments and Suggestions for Authors
Based on the contents here’s a summary addressing your question. The paper’s originality lies in its integrative approach — linking embryological mechanisms of liver development with the pathogenesis of congenital and developmental liver diseases. Based on the content of the uploaded article here’s a focused critique of the methodology and recommendations for additional controls the authors should consider:
- The methodology currently lacks detail for replication (exact search strings, date of final search, and criteria for excluding animal or molecular-only studies). Provide exact database queries and rationale for cutoffs. This would allow reproducibility and transparency.
- When discussing bile duct development or ductal plate malformation, no developmental stage controls are included (e.g., normal vs. abnormal fetal stages). Recommend inclusion of comparative embryonic stage data or references with normal development benchmarks.
In summary, while the study is methodologically solid with its prospective, controlled, and randomized design, integrating these improvements would significantly enhance.
Author Response
The authors would like to acknowledge the reviewer for its time and effort. The manuscript was changed accordingly.
- The database methodology was changed
- Embryonic data has highlighted in the bile duct development section
Reviewer 2 Report
Comments and Suggestions for Authors
In terms of content, this is a well-written paper, citing most of the relevant research on development and liver malformations. Most of what is described here is actually available in the literature. However, some citations are not precise or contain error, e.g. 95. "Expressions of p53 and inducible nitric oxide synthase in congenital choledochal cysts - PubMed. 658 Accessed 24.01.15."-instead of Zhan JH, Hu XL, Dai CJ, Niu J, Gu JQ. Expressions of p53 and inducible nitric oxide synthase in congenital choledochal cysts. Hepatobiliary Pancreat Dis Int. 2004 Feb;3(1):120-3. PMID: 14969853 or citation 91. Khandelwal, C.; Anand, U.; kumar, B.; Priyadarshi, R. N. should be capitalized. There are many such blunders in the References section.
Furthermore, for a better understanding of the developmental processes presented, the authors could have provided some kind of drawing explaining the processes described in the text. There are plenty of such drawings in the works cited, and I found it easier to understand what the authors meant when reading the original work than the review presented.
In the pathology section, I would suggest presenting the most frequent lesions in the form of a table. Furthermore, I believe that the use of the term 'diseases' is not appropriate for all the cases described. Please consider this and amend accordingly.
Congratulations, good work.
Author Response
The authors would like to thank the reviewer for its time and effort in the process. Revisions were made accordingly.
References have been reviewed.
A table was added to the pathology section, namely at the end of the discussion
Schemes explaining the process were also added as suggested
Reviewer 3 Report
Comments and Suggestions for Authors
The authors have presented a very interesting review concerning liver embryology and the relationships with histological analyses for the detection of liver diseases. The work is very interesting and also useful for the academic teaching aspect. The histological images are of excellent quality and perfectly illustrate the pathologies discussed by the authors. However, we would like to suggest to the authors some additions to this manuscript to increase its robustness and the validity of the message, especially for young students of pathological anatomy.
Major comments:
1) Increase the size of chapter 1 (introduction), adding some important details regarding the fundamental value of histology in the identification of liver diseases.
2) It is recommended to also add a chapter concerning the role of histology in liver biopsy inside the field of transplantation.
3) Adding an initial diagram illustrating the hepato-bilio-pancreatic district would enhance the appeal of this review.
Minor comments:
1) References in the text should be enclosed in square brackets.
Author Response
The authors acknowledge the reviewer for its time and effort in the process.
The manuscript was changed accordingly with the increase of chapter 1 and describing the role of histology in liver Tx. Liver biopsy was stressed mainly is vascular abnormalities, since tumors or cirrhosis are not covered by this manuscript. The role of biopsy was added in all the manuscript and not as a full chapter.
Round 2
Reviewer 3 Report
Comments and Suggestions for Authors
The authors have adequately responded to the reviewers' comments and have made significant changes to their manuscript. The work presented by the authors is now worthy of publication.